

# Research progress and mitigation strategies for pod shattering resistance in rapeseed

Li Liu[1], Hafiz Hassan Javed[1], Yue Hu[1], Yu-Qin Luo[1], Xiao Peng[2] and Yong-Cheng Wu[1]

[1] College of Agronomy, Sichuan Agricultural University, Chengdu, Chengdu, China
[2] Key Laboratory of Crop Ecophysiology and Farming System in Southwest China, Chengdu, Chengdu, China

## ABSTRACT

**Background.** Mature rapeseed pods typically shatter when harvested, resulting in approximately 8–12% yield loss. Adverse weather conditions and mechanized harvesting can diminish pod yield by up to 50%, primarily owing to delays in harvesting and mechanical collisions. The pod shatter resistance index (PSRI) assesses pod damage. Recent research focused on comparing pod shatter resistance among varieties, evaluating methods, and studying gene knockout mechanisms. However, there remains a pressing need to broaden the scope of research. In particular, it is essential to recognize that pod shatter, a complex trait, influenced by genetics, environment, agronomic practices, and harvest techniques. Future studies should integrate these factors to develop comprehensive strategies to mitigate pod shatter, enhancing rapeseed yields and agricultural mechanization. This review explores factors affecting pod shatter resistance and strategies to improve it.

**Methodology.** Scoping literature review that adhered to the methodological framework for systematic reviews was performed using search engines such as Google Scholar, Web of Science, and the Chinese National Knowledge Infrastructure. This review aimed to identify pertinent articles, which were subsequently subjected to thorough screening and evaluation. The protocol for this literature review involved the following key steps: definition of research questions, development of a search strategy, development of data extraction strategy, synthesis of the extracted data, and organization and analysis of the extracted data.

**Results.** The review presents strategies for enhancing rapeseed yield during mechanized harvesting, focusing on four key areas: (i) selecting and breeding shatter-resistant varieties using DNA markers to establish a robust germplasm resource; (ii) optimizing cultivation technologies and agronomic measures to elicit favorable interactions between compact plant-type genotypes and the environment, thereby facilitating nutrient-related regulatory mechanisms of rapeseed pods to improve pod dry weight and resistance; (iii) innovating combine header design and structure to better suit rapeseed harvesting; and (iv) providing training for operators to enhance their harvesting skills. These comprehensive measures aim to minimize yield loss, increase production efficiency.

**Conclusion.** To effectively reduce yield loss during mechanized harvesting of rapeseed, it is crucial to enhance resistance to pod shattering by addressing both internal physiological factors and external environmental conditions. This requires a holistic approach that includes genetic improvements, optimization of ecological conditions,

Corresponding author
Yong-Cheng Wu,
ycwu2002@163.com

careful cultivation management, and precise harvesting timing, along with ongoing research into traits related to machine harvesting to boost production efficiency and sustainability.

## INTRODUCTION

Rapeseed (*Brassica napus* L., AACC, 2n = 38) is a commercially important crop that yields an edible vegetable oil (*Karunarathna et al., 2020*). The structure of a rapeseed pod primarily consists of a stalk, two petals, a false diaphragm, a dehiscence zone (DZ), and a horn beak (*Mustafa et al., 2022*). The DZ, which is approximately two cell layers wide, is located between the replum and valves (*Dong & Wang, 2015*). As the fruit ripens, it undergoes a natural dehydration process within its structure. This process increases the force driving the separation between the fruit pods and the embryonic seating frame, surpassing the cohesive force holding the cells together in the separation layer. Consequently, the pods and the embryonic seating frame in this layer rupture, and the matured pods disperse along the ventral or dorsal sutures. This process, known as pod shattering, significantly diminishes the overall yield (*Guo et al., 2022*). Furthermore, the dispersion of seeds from wild rapeseed in fields can lead to increased labor costs and subsequent crop yield losses (*Liu et al., 2020*). The pod shatter resistance index (PSRI) assesses the extent of damage based on the number and severity of affected pods (*Pengfei et al., 2013*). Reportedly, the proportion of seeds lost to mechanical collisions that occur during cutting, drying, or combined harvesting can exceed 12% (*Gulden et al., 2017*). Moreover, delays in the harvest period due to adverse weather conditions, such as rainfall, can also reduce pod yield by up to 50% (*Li et al., 2021*). Therefore, further research is needed to develop more effective strategies for minimizing the breakage of the rapeseed pod.

The fragmentation of rapeseed pods is a complex problem involving multiple factors, including genetic characteristics of the variety, natural environment, cultivation techniques, and harvesting management (*Kuai et al., 2016*; *Fani et al., 2019*; *Kuai et al., 2017b*; *Rasheed et al., 2021*). Currently, research on the shattering mechanism of siliques mainly focuses on the differentiation and development process of the cleavage zone, which is regulated by several core transcription factors in the MAD-Box gene family (*He et al., 2018*; *Li et al., 2021*). The force required for the shattering of the siliqua primarily originates from the close association with lignification of the fruit petal edge and endocarp-*b* (*enb*) (*Geng et al., 2022a*). Moreover, fruit pod shattering is an inherent cellular separation phenomenon that leads to the inevitable softening and degradation of the cell wall (*Ferrándiz, 2002*). During ripening and dehydration, the differential shrinkage of cell walls in the lignified layer and endocarp-*b*, compared to that in the non-lignified cells of the pericarp (endocarp-a), generates internal tension, leading to cell separation facilitated by hydrolytic enzymes (*Zaman et al., 2019*). Simultaneously, polygalacturonase and cellulase further facilitated

the separation of the cellular layers by weakening intercellular connections (*Yu et al., 2020*). Despite advances in understanding the role of the DZ in fruit pod shattering, this complex phenomenon is also influenced by various other key factors, including fruit characteristics, changes in plant hormones, and various cultivation measures taken during the planting process. Unfortunately, these important factors have not been fully explored and validated in current research. Our limited knowledge of their complex interplay hinders a comprehensive grasp of fruit shatter and effective solutions. Future research should expand to investigate these areas, exploring their specific effects and interaction mechanisms to devise strategies that mitigate pod shatter and promote sustainable agriculture.

To address the pressing challenges of enhancing shatter resistance in rapeseed crops, prolonging the harvest period, and minimizing crack-induced losses, significant advancements are imperative. Given the urgent need to reduce the continuously rising labor costs and solve the problem of labor shortage, the widespread popularity and application of rapeseed combine harvesters have become a key measure to alleviate these issues (*Yin & Wang, 2012*). Recent research has made valuable contributions by analyzing the differential effects of individual pod characteristics across different varieties, comparing methods used to evaluate pod fragmentation resistance in different pods, and studying the molecular mechanisms of partial gene knockout (*Qing et al., 2021*; *Kuai et al., 2016*; *Tan et al., 2024*). Despite advancements, there are still gaps in our comprehensive understanding and enhancement of pod fragmentation resistance. A critical area that requires further exploration is the integration of multifactorial methods to account for the intricate interplay among genetic, environmental, and agronomic factors that influence this complex trait. Current research is often limited to a single variable and neglects the holistic perspective. To develop strategies to alleviate pod cracking, a comprehensive and systematic approach must be taken, with in-depth research on subtle changes in internal mechanisms and close monitoring of external environmental changes, in order to find the optimal balance between genetic improvement, cultivation management, and mechanized harvesting techniques. Therefore, this review aims to bridge this gap by investigating the multifaceted factors that affect shatter resistance in rapeseed pods and exploring strategies to bolster this resistance. By improving the resilience of pods against shattering, we can effectively reduce harvest-time yield losses, ensuring increased rapeseed production and further advancing the level of agricultural mechanization in rapeseed farming.

## SURVEY METHODOLOGY

The scoping literature review conducted in this study adhered to the methodological framework outlined by *Munn et al. (2018)*. We employed various search engines, such as Google Scholar, Web of Science, and the Chinese National Knowledge Infrastructure, to identify pertinent articles, which were subsequently subjected to thorough screening and evaluation. Our primary aim was to provide a comprehensive overview of the current understanding of seed loss attributable to pod dehiscence, its repercussions on yield and profitability, and its ramifications for the mechanized harvesting of rapeseed. This article is beneficial to a broad audience involved in agriculture, plant physiology, and genetics.

We explored the significance of crack resistance in rapeseed pods for augmenting yield and revealed methodologies for enhancing crack resistance through adaptations to mechanized harvesting, cultivation techniques, and advanced agronomic measures. Additionally, we summarized the challenges posed by limited germplasm resources in developing shatter-resistant rapeseed varieties. We reviewed the status and progress of research on modern breeding techniques, cultivation technologies, and agronomic measures. To ensure a rigorous and comprehensive review, we employed inclusion and exclusion criteria to select the most relevant articles. The selected articles were then classified and analyzed based on their relevance to research questions. We also described the metrics used in this review. The protocol for this literature review involved the following key steps:

1. Definition of research questions: Research questions guiding this systematic literature review were as follows: (1) What research approaches have been proposed to mitigate pod shatter resistance in rapeseed? (2) What are the current research achievements and future research directions in this field?

2. Development of a search strategy: We constructed four main query strings to identify relevant articles. These strings incorporated various spellings of "rapeseed", "pod shatter resistance", "agronomic measures", and "mechanized harvesting". We also included terms related to "pod length", "pod width", "number of seeds per pod", and "plant characteristics". To refine our search terms and ensure alignment with the research questions, we added another search string, focusing on keywords such as "density", "fertilizer", "plant hormone", and "gene editing".

3. Development of data extraction strategy: Data extraction process in this study involved several key steps. Initially, all relevant articles were downloaded and organized alphabetically, following the compilation of a reference list. Subsequently, a thorough screening process removed five duplicate articles. Thereafter, each article underwent comprehensive review and analysis, focusing on titles, abstracts, and conclusions. This meticulous process was conducted by all coauthors to ensure consistency and rigor in data extraction. Our objective was to extract and synthesize the most pertinent information from the literature, enabling a comprehensive understanding of the topic.

4. Synthesis of extracted data: Each coauthor independently assessed each article for relevance and quality, with meticulous attention to the research methods and main findings. Discrepancies were resolved through in-depth discussion and consensus. To ensure a comprehensive and unbiased review, we included studies with diverse methodologies and perspectives in our literature review. This approach allowed us to capture a wide range of viewpoints and minimize the potential impact of the limitations or biases of any single study.

5. Organization and analysis of extracted data: We structured the information into smaller groups and conducted a detailed descriptive analysis of the research findings. This method ensured comprehensive and meticulous examination of the data, enabling us to identify primary research topics, trends, and gaps within the field. We also summarized and analyzed the main findings in the included literature and discussed the current research status, shortcomings, and future research directions in this field.

## SELECTION CRITERIA FOR ELIGIBLE STUDIES

The original search identified 765 articles. After removing duplicates, 239 articles were identified for screening, After the removal of duplicate studies and old articles 45, titles and abstracts were assessed. We included studies based on the framework outlined in this review and the search keywords "rapeseed", "pod spot resistance", "agronomic measures", and "mechanized harvesting", yielding a total of 139 articles were selected. From these 139 articles, we meticulously extracted key information such as research methodologies, findings, and conclusions. Additionally, we evaluated the quality and reliability of each article, giving preference to high- quality publications from the past five years to ensure our findings were both timely and reliable.

Finally, after screening, 94 full text articles were retrieved and assessed for eligibility. The study selection flow chart is presented in Fig. 1. For the meta-analyses, we included articles selected in the systematic review that focus on variety characteristics and genotype differences, agronomic traits of rapeseed pods, and environmental factors. Strategies to improve rapeseed yield and adapt to mechanized harvesting include germplasm resource screening, gene identification, and physiological and biochemical mechanism analyses to alleviate rapeseed pod shattering resistance.

## FACTORS AFFECTING RAPESEED POD SHATTERING AND MECHANIZED HARVESTING

### Genetic factors

Pod shattering variability is linked to differences in genotypes and varieties, with pod-shattering traits varying independently among four hybrids (cultivars by seeding rate) and four open-pollinated *Brassica napus* cultivars. Moreover, pod shattering occurs before harvesting, which is more closely related to genetic background than environmental factors (*Cavalieri, Lewis & Gulden, 2014*). PSRI values differ among varieties, with conventional varieties exhibiting stronger resistance to pod dehiscence than hybrid varieties (*Kuai et al., 2016*). Recent studies have indicated that senescence-regulating genes may play a crucial role in pod shatter in *B. napus*, particularly the senescence-related transcription factors (TFs) *Bna.A05ABl5* and *Bna.C03ERFIAP2- 3*. For example, *Bna.A05ABl5* accelerates the shattering phenomenon by upregulating the expression of *Bna.A10SAG2* and *Bna.C03ERFIAP2* in the DZ. Additionally, the upregulation of *Bna.C03ERFIAP2-3* may be involved in the transcription of downstream *SHP1/2* and late embryogenesis abundant proteins, potentially triggering the shattering mechanism (*Mahmood et al., 2023*).

Rapeseed, a member of the Brassicaceae family, is considerably similar in morphology and genome to the model plant *Arabidopsis thaliana*. Notably, the morphology and cracking mechanism of their two-horned fruits exhibit resemblances (*Stephenson et al., 2019*). The DZ of *A. thaliana* pods comprises separation and lignified layers (*Ballester & Ferrándiz, 2017*; *Zaman et al., 2019*). Research on Arabidopsis has revealed that the separation area located at the fruit valve's edge primarily involves four TFs: *SHATTERPROOF1 (SHP1)* and *SHP2*, which redundantly upregulate *INDEHISCENT (IND)* and *ALCATRAZ (ALC)* (*Ballester & Ferrándiz, 2017*). The TFs MAD-Box *SHP1* and *SHP2* are specifically expressed

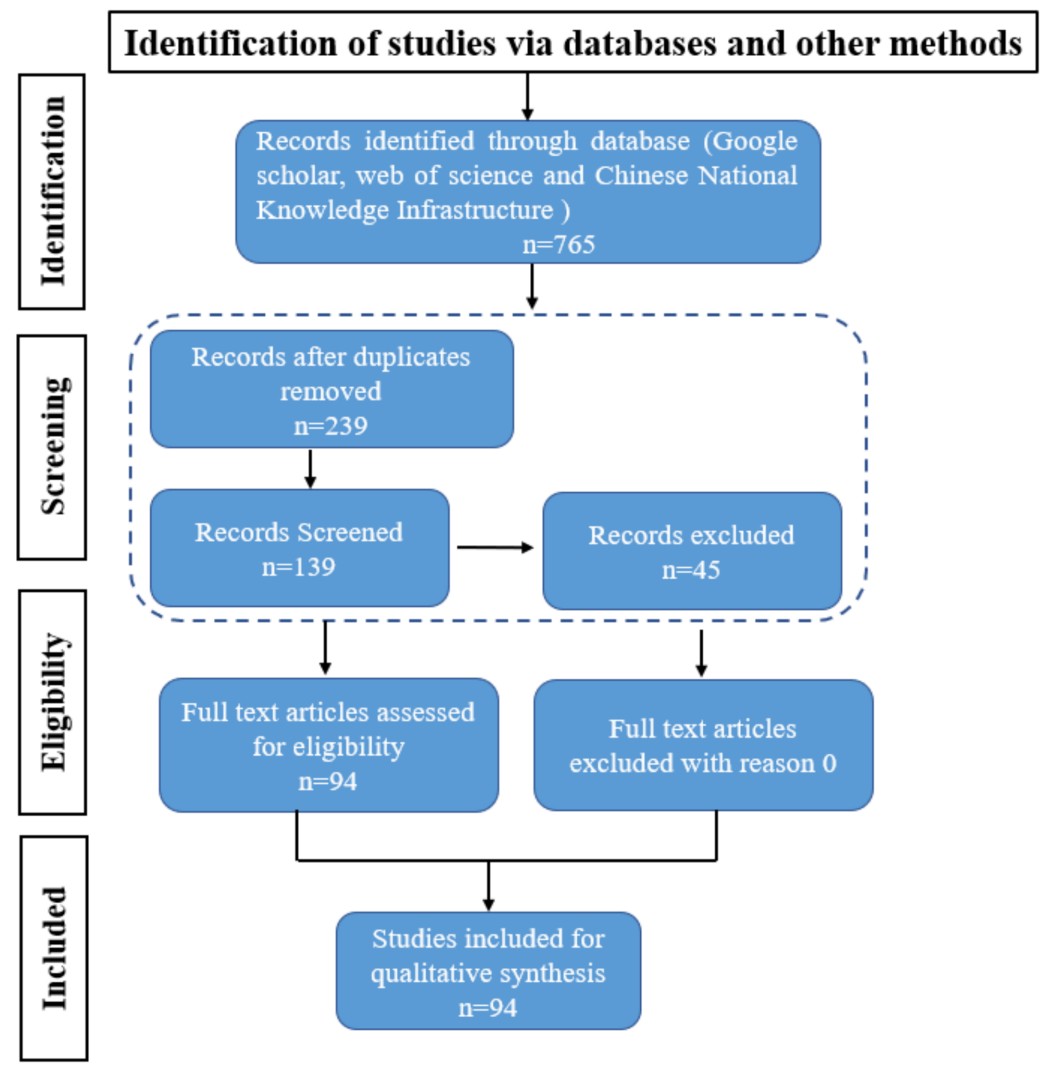

**Figure 1  PRISMA Flowchart.** The PRISMA flow diagram for the systematic review detailing the database searches, the articles number of screened.

in the off-region and positively regulate *IND* and *ALC*. Notably, individual mutations in *SHP1* or *SHP2* do not produce any phenotype, whereas double mutations in *SHP1* and *SHP2* lead to the absence of off-zone development, resulting in the formation of unsplit horn fruit (*Liljegren et al., 2000*). *IND* and *ALC* both belong to *b-HLH* TFs. *IND* controls both diferentiation of lignifed and separation layer, mutations in this gene lead to the deletion of the isolated layer and partial formation of the uncleft horn fruit. However, *ALC* mutants only show the deletion of the separation layer (*Rajani & Sundaresan, 2001*; *Liljegren et al., 2004*). The other two TFs, *FRUlTFULL* (*FUL*) and *REPLUMLESS* (*RPL*), act on the valve and the pericarp, respectively. The role of *FUL* and *RPL* is to restrict the expression of *SHP1I2* to the edge of the pericarp and regulate the normal development of the detachment region and pericarp (*Ferrandiz, Liljegren & Yanofsky, 2000*; *Roeder, Ferrandi*

& Yanofsky, 2003). Simultaneously, *SHP1I2*, *lND*, *ALC*, and *FUL* are required for the lignification of endocarp b (*ENB*) (*Ferrandiz, Liljegren & Yanofsky, 2000*). However, *NAC SECONDARY WALL THlCKENlNG PROMOTlNG FACTOR 1* (*NST1*) and *SECONDARY WALL ASSOClATED NAC DOMAlN PROTElN 1* (*SDN1)*, also known as *NST3*, play a main switch role during secondary wall lignification (*Zhong, Lee & Ye, 2010*). *NST1* is expressed in endocarp *b* and the outer lignification layer of the fruit flap, whereas *NST3* is only expressed in endocarp *b*. Mutations in *nst1* can lead to loss of the outer lignification to form anti-cleft kernels, while *nst1 nst3* double mutants exhibited a complete absence of secondary walls throughout the siliques, except for the vascular vessels, resulting siliques were indehiscent (*Mitsuda & Ohme0Takagi, 2008*). Collectively, these studies indicate that *SHP1I2*, *IND*, *ALC*, *FUL*, *NST1*, and *NST3* regulate the formation of the kernels and inner cortex, thus affecting the kernels shattering. Currently, through identification and screening of resistant germplasm resources and a comprehensive examination of the genetic control structure of the crack angle trait, research has definitively indeltifed the specific crack angle resistance gene *BnSHP1.A9*. This discovery was enabled by the detection of a highly methylated copia-like reverse transposon insertion in the upstream sequence of the *BnSHP1.A9* promoter region, effectively suppressing its expression. Consequently, this led to an enhancement in the crack angle resistance of rapeseed (*Liu et al., 2020*). Thus, conventional varieties can be used as breeding materials to develop plants resistant to pod dehiscence.

Phytohormones, also known as plant endogenous hormones, are trace organic substances produced through plant metabolism. They play a crucial role in regulating pod shattering by influencing enzyme activity or differentiation in the separation layer within the DZ. Among the five endogenous hormones—auxin, gibberellin, cytokinin, abscisic acid, and ethylene—auxin and ethylene levels exhibit close correlation with $\beta$-glucosidase activity. Reduction in auxin content in the valve edge separation zone before rapeseed pod dehydration triggers a tissue- specific increase in cellulase $\beta$-1,4-glucanase activity. This increase facilitates cell wall degradation, ultimately promoting pod shattering (*Chauvaux et al., 1997*). When auxin levels are low, cells within the cleavage zone become susceptible to ethylene, which can stimulate the synthesis of cellulase ($\beta$-1,4-glucanase) and its release into the extracellular space, leading to cell separation in the cleavage zone (*Osborne & Sargent, 1976*). The interaction between auxin and ethylene results in rapeseed pod shattering (*Child et al., 1999*). Moreover, mature rapeseed pods undergo respiratory climacteric, where the increase in ethylene content is accompanied by an increase in the activity of $\beta$-1,4-glucanase, which promote enzymatic degradation of the cell wall (*Child et al., 1999*; *Meakin & Roberts, 1991*).

In *Arabidopsis* pods, lower levels of auxin are essential for the development and formation of the separation layer, during which the auxin content of rape kernels exhibits a decreasing trend before cracking (*De Folter, 2016*). In this process, *IND*, a key regulator, directly interacts with ACG phosphokinase *PINOID* (*PID*) and *WAG2* to control the distribution of the growth hormone transporter *PIN-FORMED3* (*PIN3*). In the early stages of horn development, *IND* ensures efficient transport and accumulation of auxin from the peripheral tissue to the edge of the petal by inhibiting the expression

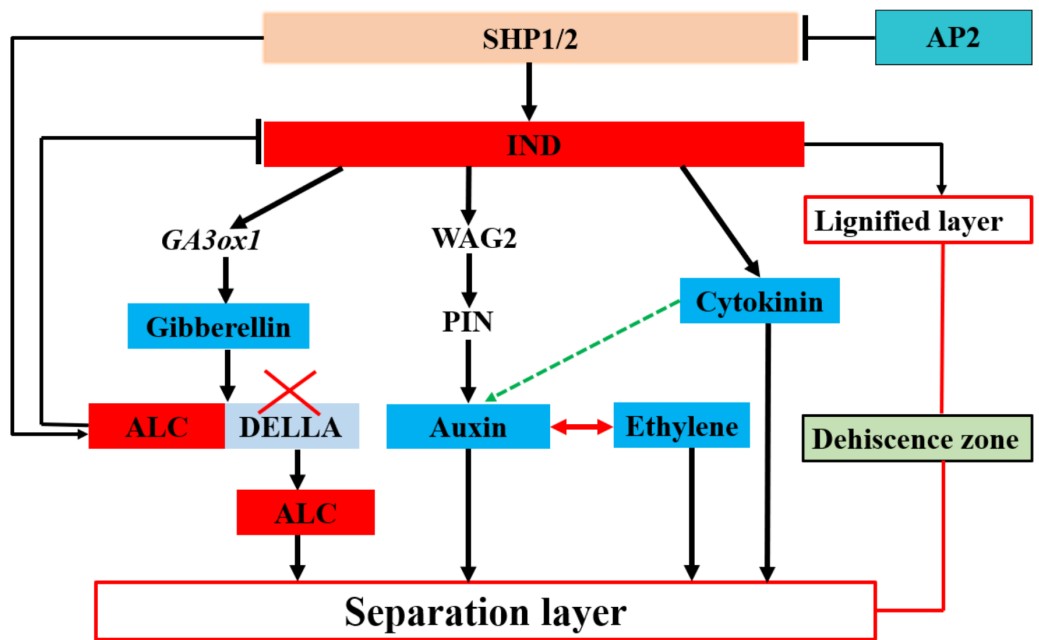

**Figure 2** **Transcription factors regulate plant hormones involved in the separation layer formation in Arabidopsis.** Plant hormones are mainly regulated by the transcription factors Indehiscent (IND) and Alcatraz (ALC), which, in turn, are regulated by the transcription factors Shatterproof1 (SHP1) and SHP2. Apetala2 (AP2) negatively regulates SHP1 and SHP2. The transcription factor IND mainly regulates gibberellin, auxin, and cytokinin. The pod dehiscence zone (DZ) comprises a separation and lignified layer. Blue indicates hormones involved in shattering; discontinuous green arrows indicate hypothetical relationships not fully supported by experimental data.

of *PID* and *WAG2* in the separation zone—a process that is essential to promote normal development of the separation zone (*Van Gelderen et al., 2016*). Simultaneously, *IND* activates the expression of *ATGA3OX1* (Fig. 2), which induces gibberellin synthesis in the DZ. Moreover, *ALC* interacts with the DELLA protein to inhibit the activation of *ALC* targets. Conversely, gibberellin synthesis triggers DELLA degradation, leading to the release of *ALC*. Subsequently, *ALC* regulates the expression of its target genes and guides the differentiation of the separation layer, ultimately causing pod shattering (*Ballester & Ferrándiz, 2017*; *Ogutcen et al., 2018*). The formation and differentiation of the *Arabidopsis* pod DZ by cytokinins (CTKs) causes pod dehiscence. Although *IND* regulates CTKs, the mechanisms underlying this regulation remain unclear (*Heim et al., 2003*). Additionally, abscisic acid (ABA) regulates pod fragmentation in *Arabidopsis* and *Brassica* plants. An increase in ABA levels induces the expression of several genes, leading to a rapid reduction in the effect elicited by ABA, which may play an essential role in pod fragmentation (*Huang et al., 2007*).

In summary, these findings indicate that plant hormones play a pivotal role in the activities of enzymes and the differentiation of the separation layer within rapeseed pods. Although extensive research has been conducted on the role of hormones in the process of rapeseed pod shattering (Fig. 2), further exploration to fully comprehend the synergistic

or antagonistic mechanisms of these hormonal signals in regulating enzyme activity and the function of the separation layer remain necessary. Moreover, while the interaction between auxin and ethylene in the cleavage region is well- documented, the potential synergistic effects of auxin and cytokinin in the pod DZ, as well as the antagonistic relationship between ABA and auxin in the cleavage process, warrant further investigation. Plant alkaline helix-loop-helix (bHLH) TFs are also integral to numerous physiological processes and particularly crucial in non-biological stress responses (*Zuo, Lee & Kang, 2023*). IND, belonging to the non-classical category of bHLH TFs, necessitates further investigation to achieve independent regulation of the lignification and separation layers. The striking similarities in morphology and genome between rapeseed and the model plant *A. Thaliana* offer both opportunities and challenges for studying hormone interactions during separation layer differentiation in pods. Such research not only advances our understanding of fundamental plant growth principles but also offers potential avenues for innovative strategies to enhance sustainable cultivation of rapeseed. For example, by precisely regulating plant hormone levels, we may be able to optimize the development process of rapeseed pods and reduce pod cracking, thereby increasing yield and quality of rapeseed.

## Effects of pod and plant characteristics on pod shatter resistance

Morphological characteristics of pods are strongly correlated with their resistance to shattering. the capacity of rapeseed pods to withstand tension is associated with pod length, width, thickness, and volume, as well as with peel thickness and weight, beak length, number of seeds per pod, thousand-seed weight, size of the DZ and its vascular bundle, and water content (Table 1). Moreover, strong positive correlation has been observed between pod dehiscence resistance and pod length, thickness, skin thickness and weight, as well as moisture content of the pod. Varieties with an enlarged replum-valve joint area (RVJA) are more resistant to pod shattering (*Hu et al., 2015*; *Zaman et al., 2019*). Pod length is positively correlated with RVJA, and a large RVJA is associated with the anti-shatter structure observed in long pods (*Braatz et al., 2018*) (Fig. 3). The correlation between the resistance that rapeseed pods exhibit to twisting forces and the number of seeds per pod, pod position, and the width-to-thickness ratio must be investigated. Although studies on the association between the width-to-thickness ratio and shattering resistance in soybeans have been conducted, few studies have examined pod characteristics. A recent study suggested that resistance to pod dehiscence correlates with the length of the main inflorescence and the height of the branches. Specifically, branch pods exhibit significantly higher resistance to shattering compared than that of pods on the main inflorescence, with the lowest level of resistance observed at the top of the main inflorescence (*Qing et al., 2021*). A high degree of pod shatter resistance has been reported in compact varieties, with higher branch positions and plant heights not exceeding 1.6 m (*Qing et al., 2021a*). The inconsistent nature of these findings (Table 1) may be attributed to the fact that pod resistance and characteristics were analyzed in different genotypes under different experimental conditions, indicating the need for further research under comparable conditions for a reliable conclusion.

**Table 1  The correlation between the resistance to shattering and the characteristics of pod traits.**

| Serial number | Morphological characteristics of the pods | Positive correlation (reference) | Negative correlation (reference) | No significance (reference) |
|---|---|---|---|---|
| 1 | Pod length Pod width Pod thickness Pod volume Pericarp thickness Pericarp weight Thousand seed weight | *Kuai et al. (2016)*; *Summers et al. (2003)*; *Udall et al. (2006)*; *Wang, Ripley & Rakow (2007)*; *Liu et al. (2013)*; *Braatz et al. (2018)*; *Qing et al., (2021)*; *Wang et al. (2019)*; *Morgan et al. (1998)*; *Jiachen et al. (2013)* | —* | *Qing et al. (2021)*; *Morgan et al. (1998)* |
| 2 | Beak length | —* | *Morgan et al. (1998)* | *Qing et al. (2021)* |
| 3 | Grain-to-shell ratio | —* | *Huiming et al. (2013)* | —* |
| 4 | Width-to-thickness ratio | —* | —* | *Liu et al. (2022)* |
| 5 | Pod position | *Romkaew et al. (2007)* | —* | *Kadkol, Halloran & MacMillan (1985)* |
| 6 | Dehiscence zone size and size of its vascular tissue | *Child et al. (2003)* | —* | —* |
| 7 | Number of seeds per pod | *Qing et al. (2021)* | *Jiachen et al. (2013)* | *Morgan et al. (1998)* |
| 8 | Moisture content of pods | *Li et al. (2012)* | —* | —* |

**Notes.**
—*Indicates a lack of relevant records.

Rapeseed pod characteristics, such as pod shell weight and water content, significantly impact shattering resistance and may serve as valuable screening indicators for shatter resistance (*Kuai et al., 2016*). Weight and water content of the pericarp are related to the quantity and composition of carbohydrates. For example, cellulose (hemicellulose) and lignin are important components of the cell walls of pods, which can affect their mechanical properties (*Murgia et al., 2017*). High cellulose or hemicellulose content enhances the mechanical support of cells in the separation layer, ensures the toughness of the cell wall, hinders the separation of cells in the separation layer, and reduces pod loss (*Sofi et al., 2022*). The increase in lignin content in rapeseed pods can increase the level of lignification at the junction of the fruit petal and embryo seat frame and thicken the cell wall of the separation zone (*Kuai et al., 2016*). Elevated lignin and cellulose contents in the pod peel enhance the mechanical strength of the cell wall and pod shattering resistance (*Dong et al., 2017*; *Kuai et al., 2017b*). Moreover, many studies have demonstrated that the water content of rapeseed pods is the main factor affecting dehiscence resistance (Table 1), as pods lose water before dehiscence and experience dehydration stress. Moreover, reduction in water content of the lignified layer of the rapeseed pod DZ increases the mechanical strength of lignin and the decomposition of the separation layer *via* the upregulation of senescence-associated genes (*Mahmood et al., 2023*). However, *Qing et al. (2021b)* studied the water content of different varieties of pods is not the primary factor affecting dehiscence resistance Instead, the genotype of rapeseed has a greater influence on the resistance. Furthermore, for the same variety, a higher water content corresponds to a greater resistance of rapeseed pods

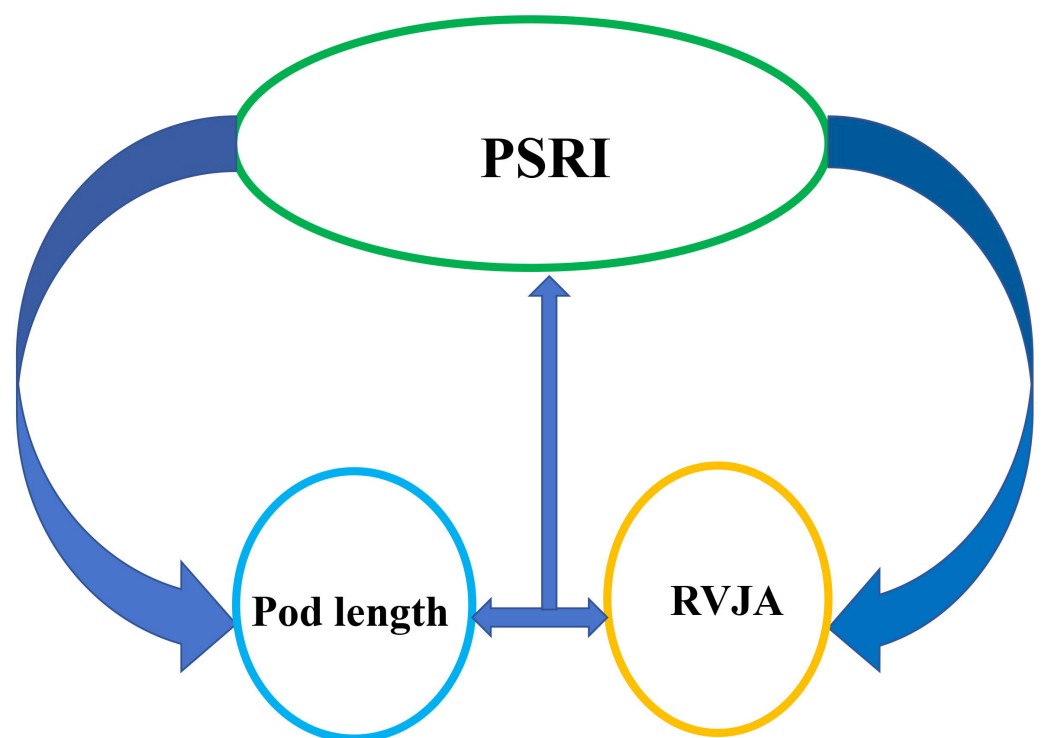

**Figure 3** **Schematic illustration of the relationship between the pod shatter resistance index (PSRI) and pod length and replum–valve joint area (RVJA).** Pod length is positively correlated with the RVJA. A large RVJA is associated with the highly anti-shatter structure in rapeseed pods. Increased pod length and enlarged RVJA show increased pod shattering resistance.

to dehiscence. The implication is that the selection of cultivars with elevated levels of shattering resistance should be prioritized in order to mitigate harvest losses.

## Environmental factors affecting seed-shattering

Under drought stress, pod shattering negatively affects the yield of rapeseed. This phenomenon may be attributed to alterations in the twisting force of the pod walls or the structural strength of the DZ in pod sutures that induces pod shattering, contributing to seed loss and reduced harvest yield (*Parker, Lo & Gepts, 2021*). During the reproductive development stage of rapeseed, drought conditions may induce the production of small and medium-sized rapeseed, leading to a decrease in thousand-seed weight (*Hatzig et al., 2018*). Moreover, the thousand-seed weight is significantly correlated with pod resistance (*Qing et al., 2021*). Therefore, a decrease in thousand-seed weight can exacerbate the shattering of siliques, ultimately significantly reducing the total yield of rapeseed. Additionally, drought conditions adversely affect various morphological attributes of plants, such as reducing moisture content, decreasing the number of branches, and diminishing biological yield (*Singh et al., 2022*). Therefore, drought stress induces reductions in both the height of rape plants and the abundance of photosynthetic pigments and branches per plant (*Raza, 2021*). Moreover, water scarcity, particularly during the initial stages of flowering in rapeseed plants, can result in a 56% decrease in the number of pods per plant

(*Fani et al., 2019*). Consequently, this adversity also leads to a significant reduction in key yield components, including the number of siliquae per plant, the number of seeds per siliqua, and the thousand-seed weight (*Lo et al., 2021*; *Liu et al., 2022*). These effects are primarily attributed to the lower photosynthetic capacity and the scarcity of photosynthetic substances resulting from water stress, which subsequently impacts the weight of siliques (*Secchi et al., 2023*; *Batool et al., 2022*).

Prior investigations have established a strong correlation between the silique dry weight and silique shatter index (*Kuai et al., 2017a*). Consequently, a reduction in silique dry weight and an increase in the shatter rate of siliques lead to a decrease in overall yield. Furthermore, extreme temperatures significantly disrupt the persistence, growth characteristics, and physiological advancements of rapeseed (*Elferjani & Soolanayakanahally, 2018*). Temperature stress can induce alterations in photosynthesis and other metabolic processes, which decrease solar radiation absorption and carbon acclimatization throughout the plant's life cycle, ultimately hindering its growth and development (*Raza, 2021*). Higher temperatures can reduce the pod's torsion force within its inner sclerenchyma, ultimately leading to pod shattering and yield loss (*Jia et al., 2021*). Silique shattering can occur during or before harvest, representing two distinct types of damage. Under favorable weather conditions, seed loss before harvesting may be as high as 2.5%, whereas under adverse weather conditions, such as changes in temperature and humidity, seed loss due to shatter may be as high as >12% or >400 kg ha$^{-1}$ (*Ahmad et al., 2022*). Moreover, factors such as low humidity, high temperature, rapid temperature changes, wetting, and drying, can reduce the seed/pod moisture content, potentially triggering pod shattering in rapeseed plants (*Maity et al., 2021*). Thus, gaining deeper insights into the correlation between pod shatter resistance and environmental factors could mitigate yield losses, facilitate the development of adaptable cultivars, and ensure a consistent supply of edible rape oil amid shifting climatic conditions.

## MITIGATION STRATEGIES TO IMPROVE POD SHATTERING RESISTANCE IN RAPESEED

### Breeding strategies to enhance pod shattering resistance

Screening and identifying rapeseed germplasm resources and analyzing the genetic basis of shattering resistance have led to the identification of two genes that regulate the resistance of pods to shattering. *Liu et al. (2020)* identified the shattering-resistance gene *BnSHP1.A9*. Specifically, the insertion of a highly methylated copia-like retrotransposon in the upstream sequence of the promoter region of *BnSHP1.A9* inhibited its expression, leading to enhanced shattering resistance in rapeseed. Future manipulations of this gene, including the introduction of further mutations *via* gene editing and knockout of homologous genes, may improve resistance to pod shattering and reduce yield losses in rapeseed. *Chu et al. (2022)* identified a rapeseed line, OR88, with a highly shatter-resistant structure in which the shatter zone harbors a lignified layer bridge (LLB), and the LLB structure differentiates at the 12th stage of pistil development. Genetic analyses have indicated that the LLB is regulated by a single recessive gene (*BnTCP.C09*), which regulates pod shatter resistance.

In the future, genes regulating the formation of the abscission-lignified layer adhesion structure in this manner could be transferred to desired varieties of *B. napus via* a molecular marker-assisted selection process. This strategy may lead to a considerable improvement in the dehiscence resistance of rapeseed varieties and accelerate the mechanization of rapeseed production.

## Application of gene editing technology to enhance pod shatter resistance

The clustered regularly interspaced short palindromic repeats (CRISPR)/associated protein 9 (Cas9) system offers a versatile tool for targeted gene modification, addressing concerns related to gene redundancy, manipulating gene expression, and acquiring desired traits. Its applications encompass a range of processes, such as selecting target genes based on desired traits, generating single or multiple copy knockouts, designing and synthesizing sgRNA, constructing CRISPR/Cas9 plasmids, performing transformation, screening for target gene editing in rapeseed, and analyzing mutant phenotypes (*Tian et al., 2022*). Considering that pod splitting involves a cell separation process, successful editing of the plant small peptide gene, *BnaIDA- A07IC06*, using the CRISPR/Cas9 system has prevented the shedding of floral organs, reduced the incidence of *Sclerotinia* infection, and improved dehiscence resistance of the rapeseed pods (*Geng et al., 2022b*). A previous study (*Zaman et al., 2019*), which found that the *Arabidopsis* gene *JAGGED* (*JAG*) plays a key role in the fruit-shattering control network, used the CRISPR/Cas9 system to perform multiplex genome editing of five homologous *JAG* genes (*BnJAG.A02*, *BnJAG.C02*, *BnJAG.C06*, *BnJAG.A07*, and *BnJAG.A08*). The resulting whole pod was shaped like a "callus" without any obvious structures, such as fruit, petal, or septum, and did not undergo abscission. Additionally, the septum was not completely covered by the cuticle, and pseudo-seeds were formed on both sides of the septum. However, these seeds did not develop or mature. When only the *BnJAG.A08*-NUBBIN-Like (*BnJAG.A08*) mutation was present, the septum was well-developed, entirely covered with the cuticle, and displayed two well-developed valves; however, the whole pod was shorter and thicker than the wild-type pod. The increased anti-dehiscence capacity of shortened and thickened pods could be attributed to the small amounts of lignin deposited in the lignified layer, concave and convex shapes of fruits, shortened pod length, and enlarged RVJA (*Ding et al., 2015*). Moreover, two homologs of *BnALC* (BnaA07g12110D and BnaC07g16290D) were simultaneously edited using CRISPR/Cas9 technology, and a genetically stable pod-shatter-resistant mutant was obtained (*Braatz et al., 2017*). The shattering process of rapeseed has been validated to be governed by eight genes in *Arabidopsis*, encompassing *SHP1*, *SHP2*, *FUL*, *IND*, *ALC*, *NAC*, *RPL*, and polygalacturonase (*PG*) (*Afridi et al., 2022*). Therefore, advanced gene editing technology offers the potential to cultivate rapeseed pod varieties with shatter resistance.

## OPTIMIZATION OF CULTIVATION TECHNIQUES AND AGRONOMIC MEASURES

### Planting density

The pod shattering resistance and yield of different rapeseed varieties vary with planting density. For the Zhongshuang 11 variety, the shattering resistance index gradually decreases with increasing planting density, reaching its maximum value at a density of $1.5 \times 10^5$ plants ha$^{-1}$. In the Huayouza 9 variety, the shattering resistance index first increases and then decreases with increasing density, with the maximum index value observed at a density of $3.0 \times 10^5$ plants ha$^{-1}$ (*Xiao-yong et al., 2018*). At high plant density ($4.5 \times 10^5$ plants ha$^{-1}$), plant structure is modified to reduce nitrogen demand (*Li et al., 2014*). This strategy improves nitrogen utilization efficiency, allowing the pod to optimally utilize light energy, which enhances the photosynthetic efficiency and the growth and development of the pod itself. Consequently, the thickness of the pod skin increases, which in turn affects pod shattering resistance in rapeseed and facilitates the achievement of target yields. At low densities (52 plants m$^{-2}$), plants are tall and thick-stemmed, and cutting becomes challenging as the normal operation of the reel and header is inhibited. At lower densities, plants are shorter, stems are thicker, and pods are closer to the ground; this increases pod shattering. However, at medium densities (101 plants m$^{-2}$), plants have few branches and moderate height, where the presence of high branches facilitates mechanized harvesting (*Haifeng et al., 2010*). Moreover, a reasonable planting density can effectively reduce competition among individuals, regulate individual growth and conflicts across groups, and achieve maximum yield (*Cong et al., 2023*). However, excessively high planting density can result in mutual obstruction and inadequate light penetration among plants, subsequently influencing the population structure and photosynthesis, ultimately affecting the accumulation and transportation of carbohydrates between stems and pods (*Kuai et al., 2022*). This results in a gradual decrease in the effective number of pods per plant and the number of seeds per pod, thereby reducing yield (*Zheng et al., 2013*). Additionally, combining a planting density of $45 \times 10^4$ plants ha$^{-1}$ with a narrow row spacing of 15 cm in central China has resulted in decreased plant height and branch angle, which prevented pod shattering and improved pod shattering resistance (*Kuai et al., 2015*).

Leaf angle is a structural component that determines the planting density of rapeseed, affecting canopy transmittance and final plant economic yield (*Zheng et al., 2022*). The change in planting density primarily affects plant traits. With the increase in planting density, plant height and number of branches, main sequence angles, and single plant angles of rapeseed varieties significantly decrease, which in turn affects the maturity of pods and influences pod cracking resistance. Overall, appropriate planting densities enhance crack resistance and provide high yield. However, determining the appropriate planting density is a multidimensional process that involves integrating various factors, including variety characteristics (such as growth period, growth habits, plant type characteristics, and adaptability), ecological conditions (water availability, soil fertility, light intensity), and agricultural practices (sowing time, field management convenience, and harvesting conditions). By adjusting planting density based on the growth needs of varieties and the

resource status of the fields while considering actual production management situations, we can maximize land resource utilization and optimize the crop growth environment. This approach enhances light energy utilization and production efficiency, ultimately leading to higher crop yields.

## Application of fertilizers and plant growth regulators

Pod shatter resistance has increased with the application of appropriate amounts of silicon fertilizer. At the closed canopy stage, the application of 150 mg L$^{-1}$ paclobutrazol can lead to pod dry weight increase, which can result in a notable increase in the number of pods per plant and thousand- grain weight of rapeseed, ultimately enhancing overall yield (*Kuai et al., 2017a*; *Kuai et al., 2015*). Spraying silicon fertilizer on the leaf surface can improve pod shatter resistance in rapeseed (*Ahmad et al., 2022*). Under severe drought conditions, the application of silicon to leaves leads to substantial increases in grain filling and reduces pod shattering (*Fani et al., 2019*). During the bolting stage of rapeseed (with a height of 8–12 cm), the application of monosilicic acid- liquid silicon fertilizer (Si[OH]4) (0.96 mM) to the root increases the dry weight of the pod, which is related to the pod dehiscence index and is beneficial for improving pod dehiscence resistance and yield (*Ivanov & Harizanova, 2022*). The increased SiO$_2$ content and dry weight of the pod appear to strengthen the pod by silica deposition in the cell wall of the pod, thereby reducing pod separation and seed loss. Furthermore, the lignin content increases with increasing silicon fertilizer concentration, thereby improving the shattering resistance of rapeseed and optimizing mechanical harvesting characteristics (*Kuai et al., 2017b*).

Spraying plant growth regulators, such as paclobutrazol (150 mg L$^{-1}$), during the closed canopy and early bud stages of rapeseed growth may improve plant growth and pod shatter, making these plants amenable to mechanized harvesting. The application of a suitable concentration of the plant growth regulator paclobutrazol prolongs the growth period and leads to an increase in the accumulation and distribution of biomass by affecting its maximum accumulation, net assimilation, and relative growth rates, thereby making rapeseed plants resistant to dehiscence (*Yang et al., 2015*; *Kuai et al., 2017a*). Moreover, concomitant application of high doses of ammonium sulfate (2 L ha$^{-1}$) and plant growth regulators (1.5 L ha$^{-1}$) achieves excellent results in terms of rapeseed growth, yield, oil content, thousand kernel weight (g), and water content. Furthermore, the thousand-seed weight and water content are positively correlated with indicators of dehiscence resistance (*Qing et al., 2021*; *Ivanov & Harizanova, 2022*). Spraying of pod sealants (novel acrylic- and trisiloxane-based) reduces pod shattering; the spraying of individual plots of the oilseed rape variety "Cultus two weeks before harvesting using an Amazone UF 901 sprayer markedly enhanced the economic benefit of winter rapeseed planting and reduced seed loss by 20–70%. Moreover, sealants reduce rapeseed shatter at maturity by enveloping the pods with a protective coating, which modulates water content and increases the physical strength of the pods. However, these effects depend on the variety of rapeseed, type of sealant, and climatic conditions (especially rainfall) (*Langowski et al., 2019*; *Steponavicius et al., 2019*). Therefore, adopting agronomic measures that improve the growth of rapeseed plants may help promote the adaptation of plants to mechanized harvesting.

## Design and optimization of the parameters of combine headers for rapeseed

Header and cutting structures produce strong collisions and vibrations during rapeseed harvesting. Thus, optimizing the design and parameters of the cutting structures of the existing rapeseed harvesters may help reduce seed loss. Additionally, dynamic testing and vibration analysis of the combines under consideration may be required to determine vibration frequency and intensity suppression methods (*e.g.*, using flexible materials) that may help eliminate or reduce the vibration of the header during harvesting (*Wang, Li & Qing, 2021*; *Zhan et al., 2023*). To address the challenges posed by dense and intricately cross-linked rapeseed branches, it is imperative for rapeseed cutting heads to be equipped not only with the primary cutter, drum, and screw conveyor but also with vertical cleaving knives positioned laterally. This enhanced configuration would facilitate the cutting of tangled branches in a manner that significantly reduces shatter loss, thereby enhancing the efficiency and precision of the harvesting process (*Yueliang, Yaoming & Lizhang, 2008*).

## Perspectives for research priorities

Uncertainty in various factors, particularly extreme weather conditions such as rainfall and temperature fluctuations, significantly impacts rapeseed yield. The absence of advanced harvesting machinery, coupled with adverse factors such as high-yield loss rates associated with mechanized harvesting, can lead to a decline in the growth and production of rapeseed pods. Our findings underscore the need to adapt pod shatter resistance to mechanized harvesting methods, cultivation techniques, and modern agronomic practices (Fig. 4). Emphasis should be placed on cultivating varieties that exhibit resilience to drought and frost, early maturity, and high nutrient utilization efficiency. Additionally, the development of suitable cultivation techniques and agronomic strategies that can mitigate losses arising from pod shatter is urgently needed.

Given the irremediable nature of short-term adverse factors such as unfavorable weather conditions, the rapeseed industry must prioritize enhancing yield. Optimizing fertilizer management and agronomic practices requires a thorough understanding of source–sink relationships and the distribution of photosynthetic products, including cell wall components, which are crucial for nutrient growth and seed development. This approach facilitates successful mechanized harvesting. Rapeseed, being an allotetraploid species, harbors a highly intricate genome characterized by numerous repetitive sequences. Consequently, knocking out all homologous genes may be crucial for achieving a reliable and desired phenotype. However, the presence of multiple copies of genes in rapeseed exhibiting high sequence similarities can interfere with accurate gene function analysis. CRISPR/Cas9 technology offers a potential solution to address gene redundancy issues, thereby enhancing pod shatter resistance. Although knocking out all homologous genes may not be feasible, the application of CRISPR/Cas9 technology can still lead to the acquisition of beneficial phenotypes. This technology enables accurate editing of genes closely related to the crack resistance of rapeseed pods, such as *BnJAG*, *BnSHP*, *BnIND*, and *BnALC*. This approach can enhance pod toughness, reduce pod cracking loss, and increase yield. Furthermore, through mutant generation and phenotype analysis, this technology

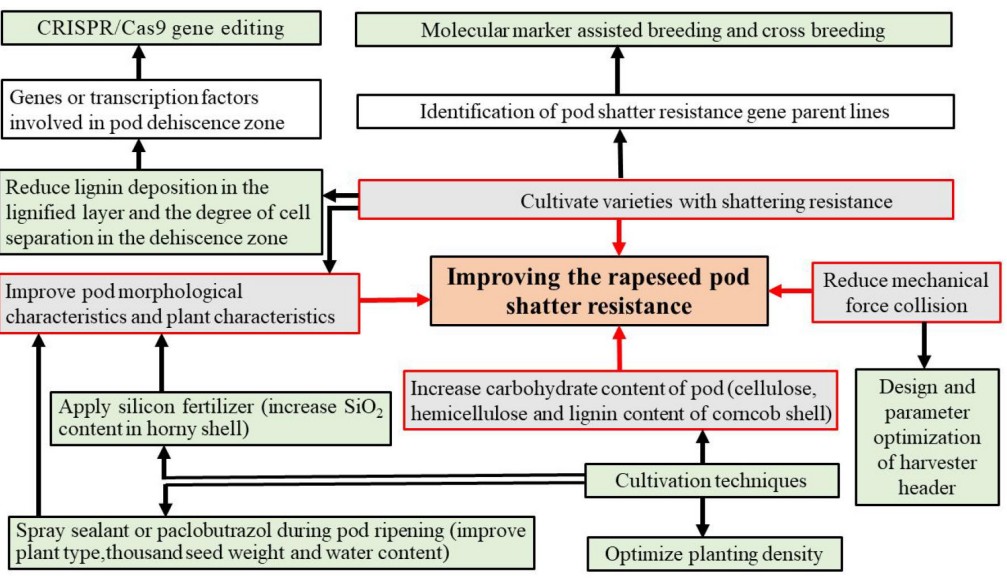

**Figure 4** **Potential strategies to manipulate pod shatter resistance.** Gray boxes indicate strategies used to manipulate pod shatter resistance; red lines link colored edges of grey boxes; while green boxes represent the methods that implement these strategies.

reveals the mechanisms of these genes in pod development and pod cracking resistance, advancing our understanding of rapeseed genetics and providing a theoretical basis for breeding. Building upon the results of gene editing and functional analysis presented in this review, researchers can identify mutants with enhanced crack resistance through screening processes. These mutants can be subsequently optimized and cultivated to develop new rapeseed varieties exhibiting robust crack resistance. The widespread planting of these new varieties will improve the stability and economic benefits of rapeseed production while promoting sustainable development within the industry.

Future studies must prioritize the utilization of shatter-resistance genes and molecular marker- assisted selection to enhance the molecular breeding of anti-shatter rapeseed varieties. These strategies should aim to strengthen pod thickness, optimize cell wall components, and facilitate the cultivation of compact plant types characterized by upright, narrow-angled leaves and smaller branch angles. This approach is expected to enhance pod dry weight, pod and seed counts, and grain weight, ultimately leading to improved pod shatter resistance and overall yield. The exploration of the intricate interplay between genotypes and environmental factors on pod shattering resistance remains crucial. Understanding how genetic traits interact with environmental variables can provide valuable insights into the development of resilient varieties that can withstand varying climatic conditions. Moreover, the mechanism underlying the tension between lignified and separation layers of the cell wall remains elusive. To address this gap, it is important to analyze the distribution of cell wall components, the interactions among hormones in the shattering region, and the differences in downstream enzyme activity and regulatory mechanisms. These approaches can provide insights into the development of

strategies to regulate the forces related to the lignification of siliques and increasing the lignin and cellulose content in the shell of the horn fruit, thereby enhancing pod integrity and resistance to shattering. On the technological front, the optimization of rapeseed combine harvesters represents a crucial area for further research. Reducing vibratory movements in the header and incorporating flexible materials into its design are crucial for minimizing pod losses during harvesting. The existence of these research gaps is primarily due to the complexity and multifaceted nature of pod shatter resistance in rapeseed. Previous studies have focused primarily on individual aspects of pod development and harvesting techniques, leaving significant knowledge gaps in the overall understanding of the phenomenon. To address these gaps, future studies must adopt a multidisciplinary approach, drawing from genetics, plant biology, agricultural engineering, and other relevant fields. By leveraging advancements in technology and knowledge, we can develop more resilient rapeseed varieties and optimized harvesting methods, contributing significantly to sustainable agriculture and food security.

## CONCLUSIONS

The yield loss during mechanized harvest can be effectively reduced by enhancing the resistance of rapeseed to pod shattering. However, the susceptibility of broken siliques is influenced by both internal physiological factors and external environmental conditions. A comprehensive understanding and in-depth analysis of these complex factors are essential prerequisites for developing effective mitigation strategies. This entails implementing a holistic approach that encompasses genetic improvement of varieties to enhance crack resistance, optimization of ecological conditions to minimize environmental stress, meticulous cultivation management to promote healthy growth, and precise harvesting timing to minimize losses. Additionally, further research on machine-harvesting-related traits, such as silique maturity, pod arrangement compactness, and silique morphology, will contribute toward enhancing production efficiency and yield in the rapeseed industry while establishing a solid foundation for its sustainable development.

## ACKNOWLEDGEMENTS

We thank the Yideji experts for their help in improving the English language and providing helpful suggestions for modifications.

### Funding

This work was supported by Biological breeding major science and technology project of Sichuan Province (2022ZDZX0015), Crop Breeding Research and Cultivation Project of Sichuan Province (2021YFYZ0005) and Sichuan Rapeseed Innovation Team of National Modern Agricultural Industrial Technology System (SCCXTD2024-03). The funders had no role in study design, data collection and analysis, decision to publish, or preparation of the manuscript.

## Grant Disclosures

The following grant information was disclosed by the authors:

Biological breeding major science and technology project of Sichuan Province: 2022ZDZX0015.

Crop Breeding Research and Cultivation Project of Sichuan Province: 2021YFYZ0005.

Sichuan Rapeseed Innovation Team of National Modern Agricultural Industrial Technology System: SCCXTD2024-03.

## Competing Interests

The authors declare no conflicts of interest.

## Author Contributions

- Li Liu conceived and designed the experiments, performed the experiments, analyzed the data, prepared figures and/or tables, authored or reviewed drafts of the article, and approved the final draft.
- Hafiz Hassan Javed analyzed the data, authored or reviewed drafts of the article, and approved the final draft.
- Yue Hu conceived and designed the experiments, analyzed the data, prepared figures and/or tables, authored or reviewed drafts of the article, and approved the final draft.
- Yu-Qin Luo performed the experiments, authored or reviewed drafts of the article, and approved the final draft.
- Xiao Peng conceived and designed the experiments, performed the experiments, analyzed the data, authored or reviewed drafts of the article, and approved the final draft.
- Yong-Cheng Wu conceived and designed the experiments, performed the experiments, analyzed the data, prepared figures and/or tables, authored or reviewed drafts of the article, and approved the final draft.

## Data Availability

This is a literature review.

## Supplemental Information

Supplemental information for this article can be found online at http://dx.doi.org/10.7717/peerj.18105#supplemental-information.

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
