# Peer review of "Research progress and mitigation strategies for pod shattering resistance in rapeseed"

_PeerJ, doi:10.7717/peerj.18105_

## Round 0.1 · original submission · Major Revisions

Dear colleagues, two experts have now reviewed your manuscript and have proposed that it has merit for publication after major revisions.

Reviewer 1 ·

Basic reporting

The manuscript gives a comprehensive analysis on targeting the factors affecting the shatter resistance of rapeseed pods and developing strategies that increase resistance to pod shattering.

Experimental design

no comment

Validity of the findings

Literature citations are detailed and accurate. But some problems need to be modified:
1.The resolution of Figure 1 is too low, please improve it.
2.As a review article, there are too few pictures attached. Please appropriately add some relevant picture descriptions of important points.
3.A total of 88 papers were cited, but 67% of them were published before 2018, and more than 50% were published 10 years ago. Even if the correctness of opinions is not shown through the publication time, as a review article, it still needs to cite more papers in recent years. It is suggested that authors should make adjustment.
4.The authors list a number of scientific ways to enhance pod shatter resistance, but is there a link between these methods and rapeseed yield? How do they affect yield?
5.The idea of the enhancement of pod shatter resistance has been proposed for many years. And the manuscript was titled “Research adapted to mechanized harvest”. But as we know, the loss of shattering pods during harvesting is still very serious in recent years and it cannot be entirely attributed to mechanization issues. Authors should give a very clear discussion of the problem.
6.“Mechanized harvesting” is one of the four main query strings to identify relevant articles. But only three papers were cited on the subject. It is hard for readers to realize that how on earth the “the research on pod shatter resistance is adapted to mechanized harvest”. The article structure is not appropriate.
7.The presentation of the “Effects of plant hormone types and contents on pod shatter” part is very confusing. The authors need to reorganize this section.

Reviewer 2 ·

Basic reporting

No comment

Experimental design

No comment

Validity of the findings

No comment

Additional comments

In this paper, the author analyzes the problem of rapeseed yield loss. Three strategies have been studied to successfully harvest rapeseed and reduce yield loss. On the one hand, varieties resistant to fragmentation were selected to establish germplasm bank, and on the other hand cultivation techniques and agronomic measures were optimized. Finally, improve the structure and design of the harvester head. This paper reviews the cultivation technology, integrated agronomic measures, design modification and parameter optimization of combine harvester for increasing rapeseed yield. The main purpose of this paper is to provide a comprehensive overview of the current understanding of seed loss due to pod cracking, the impact on yield and profitability, and the impact on mechanized harvesting of rapeseed.
The research gap in this paper is the development and improvement of threshing resistance of rapeseed Because rapeseed is prone to threshing during harvest, yield is lost, especially in the case of adverse weather conditions and mechanized harvesting. Due to the lack of crack resistant germplasm resources, the traditional genetic technology cannot completely rely on the development of crack resistant varieties. Therefore, strategies need to be proposed to ensure the successful harvesting of rapeseed using agricultural machinery and to address threshing-related challenges caused by abiotic stresses such as drought. These strategies include the selection and breeding of crack resistant varieties, the use of DNA markers to select genes for breeding, and the establishment of a systematic germplasm bank of crack resistant varieties; Optimize cultivation techniques and agronomic practices to promote favorable interactions between compact plant genotypes and the environment, thereby improving the dry weight of rapeseed; Innovate to improve the structure and design of harvesters and train operators to improve harvesting skills. Further research directions are proposed in the outlook section, including further optimization of cultivation techniques and agronomic measures, improvement of harvester head design and performance, and exploration of new gene editing and genetic improvement methods to improve threshing resistance of rapeseed. Therefore, I recommend acceptance after the author adds content, clarifies obscure content, and converges the format.
1. The description of the three steps in the development of the data extraction strategy in lines 102-106 can be a little more concise, such as deleting the second point - we deleted all duplicate papers. Add a more detailed explanation of how comprehensive and objective the data is.
2.Lines 113-115 add more detail on what constitutes a structured way to organize information and list the main research topics.
3. It is not necessary or appropriate to write seed breakage in line 123 subheadings.
4. Add details on how aging regulatory genes play a key role in pod rupture in lines 134-137.
5. Add numerical descriptions of the effects of various environmental factors in lines 242-247 as shown below.
6. Lines 281-282 explain in detail how to successfully edit genes.
7. Lines 372-375 are not easy to understand. Please explain them in detail.
8. In line 377, the conclusions and prospects of the research focus are too general and not specific. It is also suggested to strengthen the explanation of research gaps and research importance in this section.

---

## Round 0.2 · Minor Revisions

Dear colleagues, the reviewers have re-evaluated your work and decided that your manuscript has merit for publication after minor revisions

Reviewer 1 ·

Basic reporting

The paper has been revised and improved,material and methods contain a sufficient data and will be useful for other researchers to repeat. Results and discussion are correlated and gave me direct idea about the points which the authors need to touch.

Experimental design

no comment.

Validity of the findings

no comment.

Additional comments

Could you please expand the title a bit to make it more attractive for readers?

Reviewer 2 ·

Basic reporting

no comment

Experimental design

no comment

Validity of the findings

no comment

Additional comments

The article discusses the issue of pod shattering in mature rapeseed pods, leading to significant yield loss. It highlights the impact of adverse weather conditions and mechanized harvesting on pod yield and the lack of germplasm resources resistant to shattering. The study aims to investigate factors affecting shatter resistance of rapeseed pods and strategies to increase resistance, with the goal of improving rapeseed yields and promoting agricultural mechanization. The methodology involves conducting a scoping literature review following a specific methodological framework and using search engines to identify relevant articles, which are then screened and evaluated. The results of the literature review discuss cultivation technologies, agronomic measures, design modifications, and parameter optimization for combined headers to improve rapeseed yield, including the selection and cultivation of shatter-resistant varieties, optimizing cultivation technologies, and agronomic measures, and innovating the structure and design of harvesters.
One suggestion for improvement is to provide more specific information on the results of the literature review. The text briefly mentions cultivation technologies, agronomic measures, design modifications, and parameter optimization, but does not elaborate on the specific findings or key recommendations from the reviewed articles. Providing more details and examples would enhance the clarity and relevance of the results.
Additionally, it would be beneficial to include the potential implications or significance of the identified strategies in improving rapeseed yields and promoting agricultural mechanization. This could help readers understand the practical applications and potential benefits of the research findings.
Overall, the text presents a clear research objective and appropriate methodology, but can be further improved by providing more specific and detailed results, as well as discussing the practical implications of the identified strategies.
1. Introduction:Consider structuring the information into clearer sections, such as an introduction to the problem, the current research landscape, and the objectives of the present study, to improve the organization and flow of the content.
2. 301-302:mentions the impact of high temperature and drought on rapeseed yield but lacks specific details or explanations. It would be helpful to include more specific information on the mechanisms by which high temperatures and drought lead to pod shattering and other yield-reducing effects.
3. 397:briefly mentions that appropriate planting densities enhance pod crack resistance and provide high yield, but it would benefit the reader to provide more information on how to determine the appropriate planting density based on variety characteristics, ecological conditions, and agronomic practices.
4. 474-477:mentions the need to prioritize enhancing rapeseed yield and the potential of using CRISPR/Cas9 technology to achieve beneficial phenotypes. However, it would be beneficial to provide more information on the specific objectives of using CRISPR/Cas9 technology in relation to pod shatter resistance and how it could impact the rapeseed industry.
5. Present the content of Figure 2 uniformly in a horizontal layout.

---

## Round 0.3 · accepted · Accept

Dear colleagues, after assessing your revised work I am of the opinion that it has merit for publication as it stands